# The Efficacy of the Novel TSPO Ligands 2-Cl-MGV-1 and 2,4-Di-Cl-MGV-1 Compared to the Classical TSPO Ligand PK 11195 to Counteract the Release of Chemokines from LPS-Stimulated BV-2 Microglial Cells

**DOI:** 10.3390/biology9090291

**Published:** 2020-09-14

**Authors:** Sheelu Monga, Abraham Weizman, Moshe Gavish

**Affiliations:** 1Ruth and Bruce Rappaport Faculty of Medicine, Technion-Israel Institute of Technology, Haifa 31096, Israel; sheelumonga@campus.technion.ac.il; 2Sackler Faculty of Medicine, Tel Aviv University, Tel Aviv 6997801, Israel; aweizman@clalit.org.il; 3Research Unit, Geha Mental Health Center and Felsenstein Medical Research Center, Petah Tikva 4910002, Israel

**Keywords:** chemokines, CCL, IL-2, microglia, inflammation, TSPO

## Abstract

The impact of ligands of the 18 kDa translocator protein (TSPO) on the release of chemokines is not vastly investigated. In the present study, we assessed the effect of our novel TSPO ligands 2-Cl-MGV-1 and 2,4-Di-Cl-MGV-1 compared to the classical TSPO ligand PK 11195 on chemokine release in LPS-stimulated BV-2 microglial cells. As per the effect of 2-Cl-MGV-1, CCL2, CCL3, and CCL5 were inhibited by 90%, CCL8 by 97%, and IL-2 by 77% (*p* < 0.05 for all). 2,4-Di-Cl-MGV-1 inhibited CCL2 release by 92%, CCL3 by 91%, CCL5 by 90%, CCL8 by 89%, and IL-2 by 80% (*p* < 0.05 for all). PK 11195 exhibited weaker inhibitory effects: CCL2 by 22%, CCL3 by 83%, CCL5 by 34%, CCL8 by 41%, and the cytokine IL-2 by 14% (*p* < 0.05 for all). Thus, it appears that the novel TSPO ligands are potent suppressors of LPS-stimulated BV-2 microglial cells, and their inhibitory effect is larger than that of PK 11195. Such immunomodulatory effects on microglial cells may be relevant to the treatment of neurodegenerative and neuroinflammatory diseases.

## 1. Introduction

The 18 kDa translocator protein (TSPO) is a ubiquitous outer mitochondrial protein that possesses immunomodulatory effects. TSPO is highly expressed in brain astrocytes and microglia [1]. In our previous studies, we have shown the modulatory effects of the TSPO ligands 2-Cl-MGV-1, MGV-1, 2,4-Di-Cl-MGV-1, CB86, and CB204 on LPS-induced BV-2 microglial cell line activated by lipopolysaccharide (LPS) [2,3]. These TSPO ligands differ in their affinity to the TSPO [3].

Microglia are the resident macrophages of the brain and have a major role in in the host’s defense mechanism, neuroprotection, and regeneration [4]. Microglial activation starts with the detection of pathogens by Toll- like receptors (TLRs), which initiate immune signaling cascades. Microglia and astrocytes present various TLRs e.g., TLR-4, which is activated by LPS, an endotoxin produced by Gram-negative bacteria, and associated with initiation of inflammatory processes. Exposure to LPS induces systemic inflammatory response syndrome and sepsis in humans and animals [5]. Such LPS stimulation leads to the release of several cytokines, chemokines, and other inflammatory proteins. The cytokine IL-2 has multiple effects on the activation and regulation of immune responses. IL-2 stimulates natural killer (NK) cells, leading to cytolytic activity when present at high levels, and IL-2 also stimulates B cells to divide and produce antibodies [6]. An another study has demonstrated that the IL-2Rγ-JAK3 pathway for fibroblast signaling leads to increase in production of monocyte chemoattractant factors, including monocyte chemoattractant protein-1 (MCP-1) and intracellular adhesion molecule-1 (ICAM-1) [7].

Chemokines are small proteins with molecular masses of ~7–12 kDa that belong to the family of chemotactic cytokines. Chemokines regulate the cell positioning and cell recruitment into tissues, playing a pivotal role in embryogenesis, tissue development, and immune response [8]. Approximately 50 chemokines and 20 chemokine receptors have been discovered so far, and, besides their well-characterized functions in immune cell migration and inflammation, they also play a critical role in tumor initiation, promotion, and progression [9]. The CC chemokine receptors CCR2 and CCR5 as well as their cognate ligands (e.g., CCL2, CCL7 and CCL8 for CCR2, and CCL5, CCL3 or CCL4 for CCR5) have been shown to modulate monocyte/macrophage recruitment in multiple inflammatory diseases [10]. CC-chemokine ligand 2 (CCL2), also known as monocyte chemotactic protein 1, is a small pro-inflammatory chemokine and is a highly potent chemoattractant for monocytes and macrophages to sites of tissue with injury and inflammation [11]. CCL chemokine ligand 3 (CCL3), also known as macrophage inflammatory protein-1α (MIP-1α) involves in inflammatory parasitic activity [12]. Stimulation of BV-2 microglial cells by LPS leads to CCL3 release following Akt phosphorylation [13]. CCL5 is regulated, expressed, and secreted upon the activation of T cells. CCL5 at high concentrations, stimulates the production of interferon (IFN)-γ from T cells through a tyrosine kinase pathway [14]. CCL8 is another CC chemokine of the monocyte chemoattractant protein (MCP) family and is an established marker of inflammation [15]. In our previous studies, we investigated the effect of TSPO ligands on the release of pro-inflammatory cytokines and inflammatory markers from LPS-activated BV-2 microglial cells [2,3,16]. In the present study, we focused on the in vitro activity of TSPO ligands in LPS-induced microglial release of chemokines and the immune regulator cytokine IL-2, from LPS-activated BV-2 microglial cells, as a cellular model for neuro-inflammation.

## 2. Methods

### 2.1. BV-2 Microglial Cells

LPS stimulation of microglial BV-2 cells, derived from raf/myc-immortalized murine neonatal microglia was used as in vitro cellular model to study chemokines/IL-2 release (a gift from Professor Zvi Vogel laboratory, Weizmann Institute of Science, Rhovot, Israel). These cells are generally used in pharmacological, immunological, and phagocytotic studies. LPS induces microglial activation that eventually leads to the release of several pro-inflammatory cytokines, chemokines, and intracellular inflammatory proteins [17].

The BV-2 microglial cells were cultured at 37 °C with 5% CO_2_ and 90% relative humidity. Cells were incubated in Dulbecco’s modified Eagle’s medium (DMEM) containing 4.5 g/L glucose and 4 mM L-glutamine and supplemented with 5% fetal bovine serum, penicillin (100 U/mL), and streptomycin (100 µg/mL) [2,3].

### 2.2. LPS Exposure

BV-2 cells were seeded in 24-well plate and then incubated for 72 h and followed exposed to 100 ng/mL of LPS for 24 h [2,3]. The 24-well plate was centrifuged at 210 g at 4 °C for 5 min and the supernatant was collected, aliquoted, and stored in −20 °C until assayed.

### 2.3. TSPO Ligands Treatment

The synthesis of our novel TSPO ligands 2-Cl-MGV-1 and 2,4-Di-Cl-MGV-1 was described in our previous publications [3,18]. The simultaneous treatment includes the exposure of BV-2 cells to 100 ng/mL LPS with or without our novel TSPO ligands 2-Cl-MGV-1 and 2,4-Di-Cl-MGV-1 compared to the classical TSPO ligand PK 11,195 (25 µM each) for 24 h. This concentration was chosen since it was found to be the optimal concentration in several previous dose-dependent analyses [2,3,16]. The cell culture supernatant was collected and stored at −20 °C until assayed.

### 2.4. Trypan Blue Staining for Cell Counting

The cells were scrapped off before seeding into a 24-well plate. This procedure includes 1:1 ratio of cells and trypan blue dye in an Eppendorf tube. Ten microliter of this mixture was loaded into TC20 cell counting slide and then inserted into the TC20 cell counter (Bio-Rad Laboratories Ltd., Ramat Gan, Israel). The instrument automatically detects the presence of trypan blue dye and starts counting. Gating of 4–10 µm was applied for cell counting. The number of live cells was recorded on the instrument and 2.5 × 10^4^ cells were seeded per well.

### 2.5. Enzyme-Linked Immunosorbent Assay (ELISA)

Using specific ELISA kits, the levels of CCL2 (MCP1) (ab208979), CCL3 (MIP1α) (ab200017), CCL5 (RANTES) (ab215537), CCL8 (MCP2) (ab203366), and IL-2 (ab223588) were assessed. [Abcam, Zotal Ltd., Tel Aviv, Israel]. All the samples were diluted according to the manufacturer’s instructions and were compared between LPS-exposed cells and LPS + ligand-treated cells. All the reagents provided in the kit were ready for use and stored according to the manufacturer’s instructions.

*Sample preparation*: Cell culture supernatants were collected and centrifuged at 210 *g* at 4 °C for 5 min to settle the floating cells. Supernatants were carefully collected, aliquoted, and stored at −20 °C until assayed.

*Plate preparation and standards*: All the ELISA kits were provided as ready to use plates and wells. Samples were defrosted (50 µL), the antibody cocktail was diluted according to the manufacturer’s instructions. Total of 50 µL of standards, all samples, and then diluted antibody were added to the appropriate wells. The wells were sealed and incubated for 1 h at room temperature on a plate shaker. Each plate was washed with 3 × 350 µL wash buffer PT. TMB substrate (100 µL) was added and samples were incubated for 15 min in the dark on a plate shaker. Total of 100 µL of stop solution was added and mixed on a shaker for 1 min to stop the reaction. Optical density (O.D.) was recorded at 450 nm with endpoint reading.

### 2.6. Statistical Analyses

Results are presented as mean ± standard deviation (SD) or percentage. One-way analysis of variance (ANOVA) test was used as appropriate, including Bonferroni’s post-hoc test. Statistical significance was defined by *p* value < 0.05.

## 3. Results

As described previously [2] there was no effect of the vehicle (1% ethanol) on the release of cytokines and other intracellular inflammatory proteins from BV-2 cells. In the present study, we also assessed the difference between naïve and vehicle in each chemokine/IL-2 experiment. BV-2 cells were seeded in 24 well plate and incubated for 72 h in complete medium (5% FCS). Then the cells were exposed to 100 ng/mL of LPS for 24 h in starvation medium (0.5% FCS) with or without our novel TSPO ligands 2-Cl-MGV-1and 2,4-Di-Cl-MGV-1 compared to the classical TSPO ligand PK 11,195 (25 µM each). This was the common procedure for all the experiments. In all the experiments, the three TSPO ligands did not affect the chemokine/IL-2 release by themselves.

### 3.1. CCL2 (MCP1)

As shown in Figure 1, there was no significant difference between naïve and vehicle. The levels of CCL2 were robustly increased following exposure to LPS and this elevation was inhibited by 2-Cl-MGV-1 and 2,4-Di-Cl-MGV-1 up to the control levels. The classical TSPO ligand PK 11,195 was unable to inhibit the LPS-induced CCL2 elevation.

### 3.2. CCL3 (MIP1α)

As shown in Figure 2, there was no significant difference between naïve and vehicle, but the levels of CCL3 increased robustly following exposure to LPS and the release of CCL3 was inhibited completely by 2-Cl-MGV-1 and 2,4-Di-Cl-MGV-1. The classical TSPO ligand PK 11,195 also inhibited the release of CCL3 levels up to control levels.

### 3.3. CCL5 (RANTES)

As shown in Figure 3, the levels of CCL5 were increased significantly in the LPS group and this increase was suppressed by 2-Cl-MGV-1 and 2,4-Di-Cl-MGV-1 up to the control levels. The classical TSPO ligand PK 11,195 also inhibited the LPS-induced CCL5 elevation but to a lower extent than our two TSPO ligands.

### 3.4. CCL8 (MCP2)

As shown in Figure 4, the levels of CCL8 increases robustly in the LPS group and that increment was suppressed by 2-Cl-MGV-1 and 2,4-Di-Cl-MGV-1 up to the control levels. The classical TSPO ligand PK 11,195 had much weaker inhibitory effect on CCL8 elevation compared to our two TSPO ligands 2-Cl-MGV-1 and 2,4-Di-Cl-MGV-1.

### 3.5. IL-2 Cytokine 

IL-2 is one of the important cytokines in the regulation inflammatory processes. Similar to the other experiments, BV-2 cells were exposed to 100 ng/mL LPS with or without our novel TSPO ligands compared to PK 11195. ELISA was performed to assess the IL-2 levels. A marked increase in IL-2 was observed in LPS group (Figure 5). 2-Cl-MGV-1 and 2,4-Di-Cl-MGV-1 suppressed the stimulated release of IL-2 up to the control levels, but the classical TSPO ligand PK 11,195 was unable to inhibit significantly the release of LPS-stimulated IL-2 levels.

## 4. Discussion

CC chemokines, which targets primarily neutrophils rather than mononuclear cells, mediate pro-inflammatory mechanisms [19]. There are many inflammatory mediators that are responsible for the initiation and maintenance of inflammation. We used LPS stimulation of BV-2 microglial cells, as an in vitro model for neuroinflammation. We have previously found the impact of the TSPO ligands on the LPS-induced release of cytokines (TNF-α, IL-1β, IL-6, IFN-γ) and intracellular inflammatory markers (COX-2, iNOS, and NF-κB etc.,) [3,16]. In the present study, we have analyzed the inhibitory effect of three TSPO ligands on the CC-chemokines release following LPS-induced microglial activation. According to a previous study, LPS causes CCL2 secretion from‘ microglia and astrocytes and CCL2 secretion or CCR2 expression during the inflammatory conditions [20]. CCL2 secretion does not cause inflammation by itself but it mediates the response to other inflammatory stimuli [21]. Previous studies show that TSPO, as well as CCL3 and CCL5, are involved in cellular inflammatory processes [22]. Memory T cells are preferentially attracted by CCL5 leading to potent chemotactic effects on IL-2 activated T cells, basophils, and eosinophils, and eventually release of histamine. The cytokine IL-2 was also determined in this study in order to assess the effect of inhibition of inflammatory responses on CCL5.

This is the first study on the effect of the TSPO ligands on the release of chemokines and during microglial activation as a neuroinflammatory cellular model. According to the current results, TSPO ligands are potent inhibitors of the secretion of CC-chemokines, in addition to their capability in the release pro-inflammatory cytokines [2,3]. According to this study, our novel low affinity TSPO ligands inhibited the release of CCL2, CCL3, CCL5, CCL8, as well as IL-2 from LPS-stimulated microglial cells, while the classical high affinity TSPO ligand PK 11,195 suppressed significantly the release of CCL3, but exhibited only modest inhibitory effects on the secretion of other chemokines. In a recent study, we found that another high affinity TSPO ligand, CB86 was a potent inhibitor of cytokine release (TNF-α) from LPS-activated BV-2 microglial cells while the effect low affinity TSPO ligand CB204 was much weaker. Thus, it seems that the anti-inflammatory activity of TSPO ligands does not relate to their affinity to the TSPO, since the immunomodulatory potency of the ligands did not correlate with the affinity to the TSPO. It is possible that TSPO-independent cellular immune pathways are involved in the immunomodulatory effects of these ligands. Notably, in our previous study we also did not find a direct correlation between immunomodulatory activity of the TSPO ligands and their affinity to the TSPO protein, supporting the notion that non-TSPO related immune pathways play a major role in the impact on cytokines and chemokines release [2,3]. Future studies should compare the efficacy of the immunomodulatory effects of the classical TSPO ligand PK 11,195 to novel TSPO ligands, such as FGIN-1-27 which may be more specific and more effective. Moreover, future studies should evaluate the neuro-protective role of the novel TSPO ligands as reflected by elevation of anti-inflammatory mediators. In future studies, we also intend to assess the ability of the novel TSPO ligands to prevent/attenuate the phagocytic activity of the microglial cells as well as that of macrophages.

## 5. Conclusions

It appears that our novel TSPO ligands are highly effective in counteracting the LPS-induced release of some pro-inflammatory cytokines (TNF-α, IL-1β, IL-6, IFN-γ) [2,3] as well as some chemokines (CCL2, CCL3, CCL5, and CCL8) (as shown in the present study). In contrast to the novel ligands, the classical TSPO ligand PK 11,195 was a weaker inhibitor of chemokine secretion, except CCL3. The role of TSPO ligands in the treatment of neuroinflammatory and neurodegenerative diseases merits further investigation in appropriate in vitro and in vivo models.

## Figures and Tables

**Figure 1 biology-09-00291-f001:**
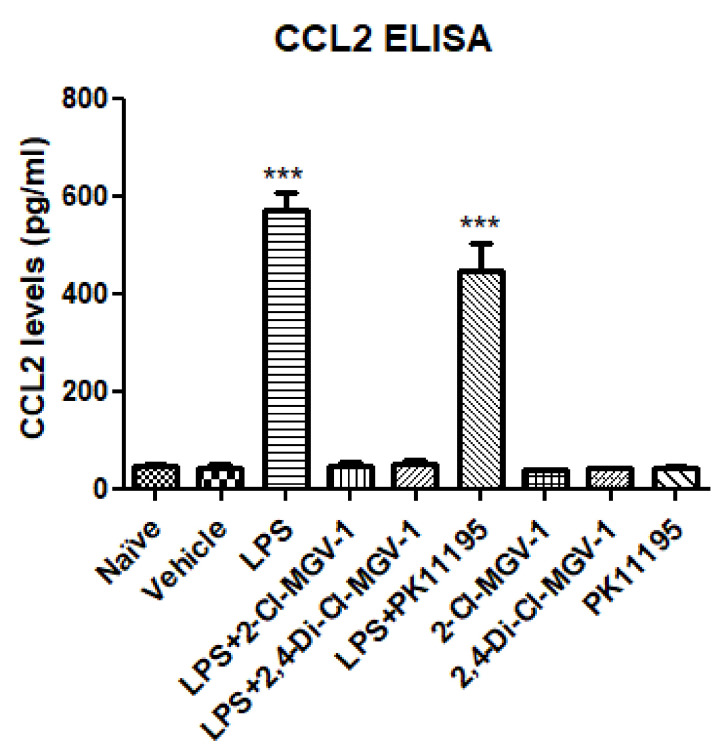
Effect of the TSPO ligands on the inflammatory chemokine CCL2. BV-2 cells were exposed to 100 ng/mL of LPS for 24 h simultaneously with or without our novel TSPO ligands compared to PK 11,195 (25 µM each). CCL2 levels (pg/mL) were calculated using a standard calibration curve and are presented as mean ± SD; four replicates in each group. ANOVA with Bonferroni’s post-hoc test was performed. *** *p* < 0.001 compared to all other groups.

**Figure 2 biology-09-00291-f002:**
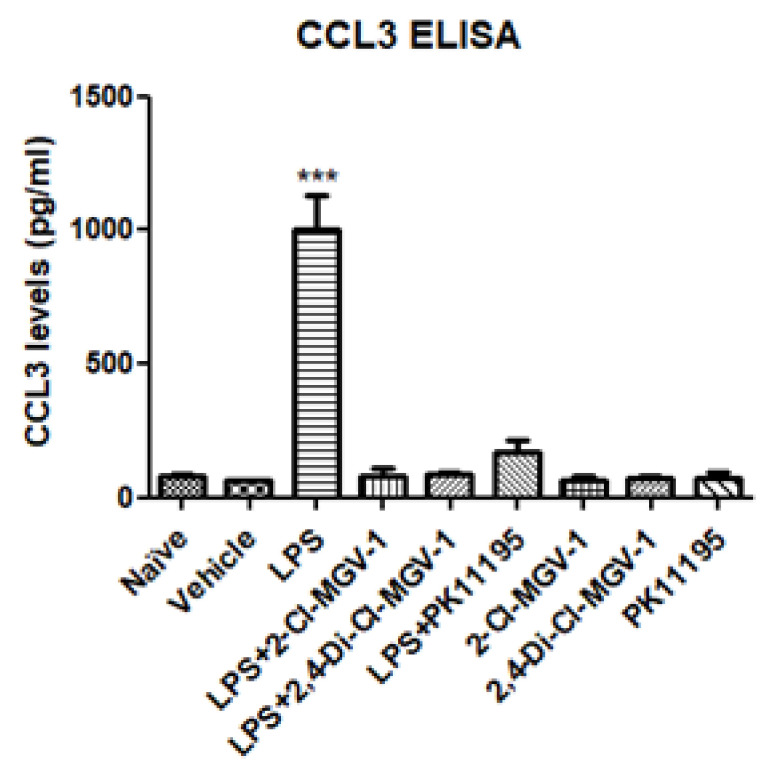
Effect of the TSPO ligands on the inflammatory chemokine CCL3. BV-2 cells were exposed to 100 ng/mL of LPS for 24 h with or without our novel TSPO ligands compared to PK 11,195 (25 µM each). CCL3 levels (pg/mL) were calculated using a standard calibration curve and are presented as mean ± SD; four replicates in each group. ANOVA followed by Bonferroni’s post-hoc test was performed. *** *p* < 0.001 compared to all other groups.

**Figure 3 biology-09-00291-f003:**
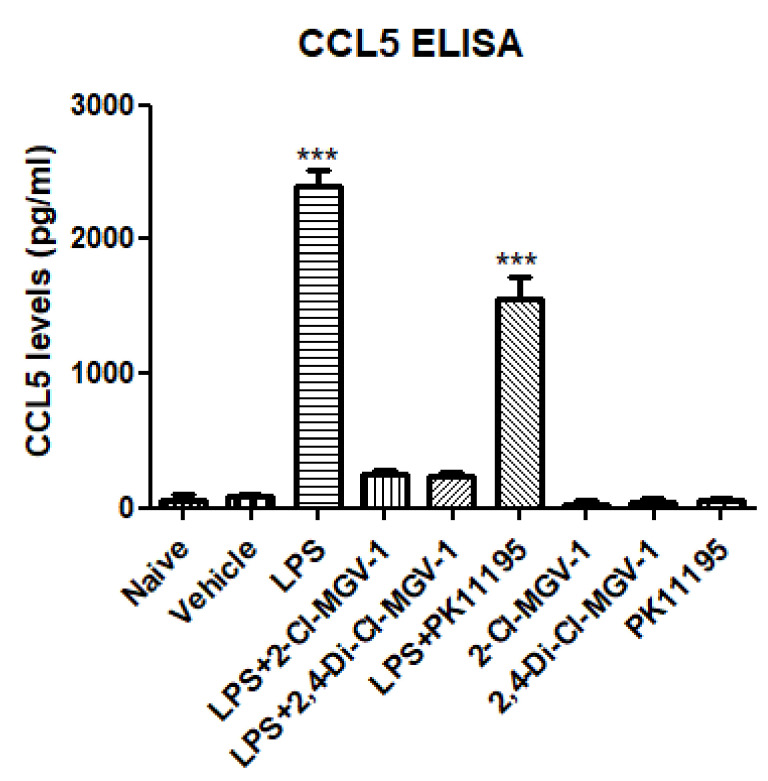
Effect of the TSPO ligands on the inflammatory chemokine CCL5. BV-2 cells were exposed to 100 ng/mL of LPS for 24 h with or without our novel TSPO ligands compared to PK 11,195 (25 µM each). CCL5 levels (pg/mL) were calculated using a standard calibration curve and are presented as mean ± SD; four replicates in each group. ANOVA followed by Bonferroni’s post-hoc test was performed. *** *p* < 0.001 compared to all other groups.

**Figure 4 biology-09-00291-f004:**
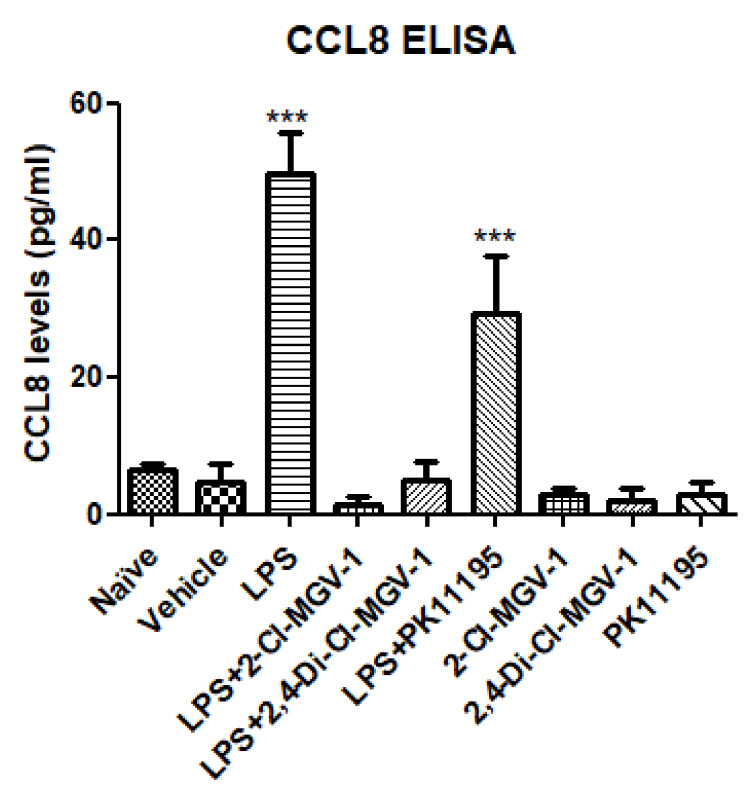
Effect of the TSPO ligands on inflammatory chemokine CCL8. BV-2 cells were exposed to 100 ng/mL of LPS for 24 h with or without our novel TSPO ligands compared to PK 11,195 (25 µM each). CCL8 levels (pg/mL) were calculated using a standard calibration curve and are presented as mean ± SD; four replicates in each. ANOVA followed by Bonferroni’s post-hoc test was performed. *** *p* < 0.001 compared to all other groups.

**Figure 5 biology-09-00291-f005:**
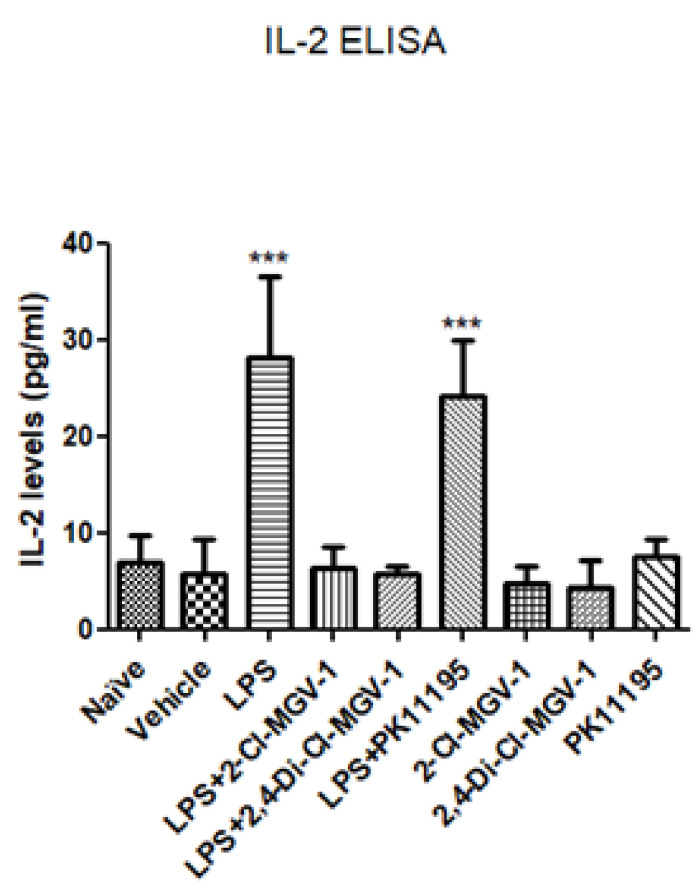
Effect of the TSPO ligands on the pro-inflammatory cytokine IL-2. BV-2 cells were exposed to 100 ng/mL of LPS for 24 h with or without our novel TSPO ligands compared to PK 11,195 (25 µM each). IL-2 levels (pg/mL) were calculated using a standard calibration curve and are presented as mean ± SD; four replicates in each. ANOVA followed by Bonferroni’s post-hoc test was performed. *** *p* < 0.001 compared to all other groups.

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
