# Peer review of "The Efficacy of the Novel TSPO Ligands 2-Cl-MGV-1 and 2,4-Di-Cl-MGV-1 Compared to the Classical TSPO Ligand PK 11195 to Counteract the Release of Chemokines from LPS-Stimulated BV-2 Microglial Cells"

_biology, 2020, doi:10.3390/biology9090291_

Round 1

Reviewer 1 Report

Increasingly, interest is focusing on roles microglial activation play in a variety of neurodegenerative diseases; as discussed by the authors, chemokine release is one of the mehanisms implicated in pathogenic effects of microglial activation.  The authors utilized an in vitro procedure to explore the ability of their two novel ligands that bind to the "18 kDa translocator protein (TSPO)" to suppress release of a variety of inflammatory chemokines; TSPO plays an important regulatory role in microglial activation and release of a variety of chemokines in response to activation by lipopolysaccharide (LPS).  Interestingly, their two novel TPSO-selective ligands were capable of suppressing release of several chemokines, whereas PK11195, the prototypic TSPO ligand, was less effective or even ineffective.  The methodology is excellent; the rationale is strong; and the results have significant translational implications.  Recently, I attended a very provocative presentation wherein a "phagocytic" role of activated microglia was presented and "discussed" as a possible pathogenic mechanism of neurodegenerative disorders.  I am wondering if the authors would consider exploring the ability of their two noel TSPO ligands to prevent or attenuate this phagocytic function in future investigations; if I recall, the presenters thought that the phagocytic activity involved activation of the complement cascade.

Author Response

We addressed the possible anti-phagocytic activity of the TSPO ligands in the final paragraph of the discussion before the conclusions.

Reviewer 2 Report

The writing in the introduction could flow better, at the moment it appears slightly disjointed, however factually correct.

Have the authors considered the use of alternative TSPO ligands such as FGIN-1-27 instead of PK11195 some authors have had more robust results using FGIN-1-27?

It would also be interesting to see if the novel ligands had a neuroprotective role increasing anti-inflammatory mediators, is this something the authors have considered? 

I would like to see further expansion oon the proposition that TSPO independent cellular pathways are involved in the immunomodulatory effects of these ligands

Author Response

We have added this notion to the discussion in the last paragraph before the conclusions. 

"It would also be interesting to see if the novel ligands had a neuro-protective role increasing anti-inflammatory mediators, is this something the authors have considered?" We related to this issue in the same paragraph in the discussion.

At present, we intend to identify immuno-modulatory pathways that are are not TSPO dependent.

Reviewer 3 Report

In this present study the authors address the question of the impact of new ligands of the 18 kDa translocator protein (TSPO) 2-Cl-MGV15 1 and 2,4-Di-Cl-MGV-1 on the release of chemokines. In this purpose the authors used LPS16 stimulated BV-2 microglial cells as previously used in their experiment and compared their results with treatment  with high affinity drug compared to the classical TSPO ligand PK11195 on chemokine release; In this context they found out that ligands induced a strong decrease of these chemokines. With this conclusion, showing an immunomodulatory effect of these new ligand is a strong reinforces TSPO as a target in neurodegenerative effects.

Nevertheless, one of the major points of these study remains on the choice of a unique one dose of the ligand. The author should make A dose dependent effect of these ligands on the chemokines release and some pharmacological studies should be done.

According to me the choice of 25 µM is very high, did the author proceed a dose dependent effect of the drug on the cells, is there any damage on the cells?

In this context of inflammation did the author check other inflammations parameters, why observe the impact of the ligands on these parameters are not analyzed in the same study

The discussion is not clear especially the sentence “Thus, it seems that the anti-inflammatory activity of TSPO ligands does not co-relate to their affinity to the 207 TSPO. It is possible that TSPO independent cellular immune pathways are involved in the 208 immunomodulatory effects of these ligands” the author must give more explanations about this part of discussion

Author Response

The reason for using one concentration of PK 11195 was added to the methods. We addressed the immono-modulatory notion in the last paragraph of the discussion before conclusions. 

Round 2

Reviewer 3 Report

The answer of the authors is in accordance with the request suggestions.

Future works should be indeed performed to study the immunomodulatory effect of the novels ligands and characterize more precisely their role in neuro protection.